# Electrophysiological correlates of focused attention on low- and high-distressed tinnitus

Rafał Milner[1][☯]*, Monika Lewandowska[2][☯], Małgorzata Ganc[1], Jan Nikadon[3], Iwona Niedziałek[4], Wiesław Wiktor Jędrzejczak[1], Henryk Skarżyński[5]

1 Department of Experimental Audiology, World Hearing Center, Institute of Physiology and Pathology of Hearing, Kajetany/Warsaw, Poland, 2 Institute of Psychology, Faculty of Philosophy and Social Sciences, Nicolaus Copernicus University, Toruń, Poland, 3 Centre for Modern Interdisciplinary Technologies, Nicolaus Copernicus University, Toruń, Poland, 4 Audiology and Phoniatrics Clinic, World Hearing Center, Institute of Physiology and Pathology of Hearing, Kajetany/Warsaw, Poland, 5 Oto-Rhino-Laryngology Surgery Clinic, World Hearing Center, Institute of Physiology and Pathology of Hearing, Kajetany/Warsaw, Poland

☯ These authors contributed equally to this work.
* rafal.milner@gmail.com

## Abstract

### Objectives

The study aimed at determining the EEG correlates of concentration on either low or high-distressed tinnitus.

### Methods

Sixty-seven patients (36 women, mean age = 50.34 ± 12.94 years) with chronic tinnitus were assigned to either a high (HD) or low (LD) tinnitus-related distress group based on THI results. All participants took part in the EEG study comprising two 3–4 min blocks of focusing on either tinnitus (Tinnitus Focus Condition, TFC) or the sensations from one's own body (Body Focus Condition, BFC). The absolute power and current density of 8 frequency bands in 7 clusters were compared between conditions and groups.

### Results

The most pronounced differences were found in the HD patients in the TFC, relative to the BFC, i.e. reduced power of frontally distributed low alpha (8–10 Hz) and posterior high alpha (10–12 Hz) as well as lower current density of 8–10 Hz rhythm over the right frontal/anterior cingulate cortex and higher middle beta (15–18 Hz) density in the precuneus. The HD, relative to LD patients, in both conditions, exhibited increased low beta (12–15 Hz) power over the left middle area and greater higher beta (15–25 Hz) power in the left posterior region.

### Conclusions

The present study contrasted bioelectrical activity, acquired when concentrating on tinnitus with EEG data collected whilst patients focused on their body. Decreased alpha power and current density in the frontal/cingulate cortex when listening to bothersome tinnitus might reflect greater cortical arousal whereas increased beta power and density in the precuneus/

**Data Availability Statement:** All data used in the analysis is available on figShare database with the accompanying DOI: https://doi.org/10.6084/m9.figshare.12643403.v1.

**Funding:** This study was supported by a grant from the Polish 1015 National Science Centre (UMO nr 2011/03/D/NZ4/02431). The funders had no role in study design, data collection and analysis, decision to publish, or preparation of the manuscript.

posterior cingulate activity in this condition could be indicative for elevated tension or augmented cognitive/emotional processing of tinnitus sound. Enhanced beta rhythm in patients with high *versus* low tinnitus distress, observed independently of the study condition, may be due to greater self-focused attention or more active processing of sensations derived from the own body.

## Introduction

Tinnitus, an auditory percept appearing in the absence of any external or internal sound source, is a serious medical and social problem affecting 10–15% of the global population [1,2]. If tinnitus becomes chronic, it can contribute to depression, anxiety, sleep disturbances, and concentration difficulties that dramatically reduce the quality of life [3–7]. Tinnitus varies in its perceptual characteristics (e.g. its laterality, type of sound, pitch, loudness), the distress it evokes (emotional reaction to its presence), comorbid conditions (e.g. hearing loss, hyperacusis, vertigo, affective disorder) and response to therapy [8]. The heterogeneity of tinnitus may explain why there is no one effective treatment for all patients and the outcomes of a given therapeutic intervention are highly variable. Patients might use various non-specific methods to cope with tinnitus including sound or music therapies [9–11], cochlear implantation [12], psychological therapies [13–19], brain stimulation [20–22], or neurofeedback training [23,24]. No effective pharmacological treatment for this condition is yet available.

Tinnitus etiologies encompass noise exposure, age-related hearing loss (presbycusis), cardiovascular and cerebrovascular disease, medications, head/neck injury, and hyper- or hypothyroidism [25]. Although the cause of tinnitus might be located outside the central nervous system, central brain structures are thought to be responsible for turning it into a chronic sensation [26]. Tinnitus has been considered as the acoustic analog of phantom pain [27,28] caused by deafferentation (incomplete afferent connection with the central nervous system) [29–32] leading to possible increases in neural network gain [33], neural plasticity [34,35], neural synchrony [36], sensitization [37] or sensory uncertainty [38]. The complementary theory proposes that deafferentation is accompanied by dysfunction of the "noise-cancellation system" involving reciprocal connections between primary auditory cortex, ventromedial prefrontal area, thalamus, and amygdala [39–41]. In the current theories, developed to integrate the abovementioned models, tinnitus has been considered as a property of multiple, parallel and partially overlapping neural networks [42] or a result of a prediction error in the auditory system [43].

Chronic tinnitus is associated with an abnormal resting-state MEG/EEG signal recorded in the auditory cortex, which could be seen as an increased activity in the range of gamma ($> 40$ Hz) [44–46], delta, theta, alpha, and beta bands [47] as well as reduced low (8–10 Hz) but not high (10–12 Hz) alpha power in the temporal area [48]. Chronic tinnitus could also manifest in resting-state EEG by lowered alpha (8–12 Hz) power, accompanied by enhanced delta activity [49]. A possible theory accounting for reduced slow-wave activity together with increased gamma over the temporal areas in tinnitus patients is the thalamo–cortical dysrhythmia model [29,45] in which tinnitus relates to persistent gamma activity in the auditory cortex due to auditory deafferentation. In the deafferented state the dominant resting-state alpha rhythm (8–12 Hz) decreases to a theta rhythm (4–7 Hz). As a result, GABA-mediated lateral inhibition is reduced, inducing gamma band activity in regions adjacent to the deafferented theta (called

an edge effect, 40). Notably, hearing loss is rarely controlled in these studies [50] but if it was, the obtained results only partially support this model.

The majority of patients are able to get used to their tinnitus percept, however, in some patients the reaction to this novel sound (tinnitus) does not reduce over time. Low-distress and high-distress tinnitus have different neural representation, as can be observed in EEG and fMRI studies on resting brain activity [50–60], when, in the absence of any external stimuli, patients are asked to relax and not think of anything in particular [61,62]. Specifically, these studies show that troublesome tinnitus is associated with increased upper beta activity in the frontal area, whereas less bothersome tinnitus is associated with enhanced delta, alpha, and lower gamma (30–40 Hz) in the bilateral temporal area [58]. In comparison to healthy controls (from a normative database), patients with bothersome tinnitus showed higher current source density in the alpha (8–12 Hz) and beta (21–32.5 Hz) bands and less synchronized delta (2–3.5 Hz) and theta (4–7.5 Hz) in the dorsal anterior cingulate cortex [59]. In the same study, high-distress tinnitus (relative to low-distress tinnitus) was associated with greater current source density in the low alpha band (8–10 Hz) in the posterior subcallosal anterior cingulate, para-hippocampal gyrus, and amygdala. Bothersome tinnitus also co-occurred with increased synchronization in the high alpha (10.5–12.5 Hz) band in the para-hippocampal gyrus, amygdala, and middle temporal gyrus; there was also decreased current source density in this band in the posterior cingulate and precuneus [59]. Previous studies have demonstrated that distress caused by tinnitus involves the global fronto-parieto-cingulate network [63,64] and is related to the alpha and beta bands in the anterior cingulate and ventromedial prefrontal/orbitofrontal cortices [54,56,59,65] or gamma activity in the auditory cortex or insula [46,66,67]. The subcallosal/ventromedial prefrontal area and the amygdala are considered as the structures forming the "noise-cancellation" system thought to underlie long-term habituation to tinnitus [40]. Most of the above structures (the anterior cingulate and prefrontal cortices, the insula) are thought to be parts of the "pain neuromatrix" [68] suggesting a common "distress network" for tinnitus and pain.

It has been also demonstrated that tinnitus-related distress (a temporary feeling) and depression (a more prolonged state) are represented in different neural networks [56,58]. Joos et al. [56] found a lateralization effect when looking for differences in spontaneous bioelectrical activity in these two states. Specifically, the authors found that distress caused by tinnitus was associated with the alpha (10.5–12.5 Hz) and beta (13–21 Hz) bands in the right frontal areas as well as beta activity (18.5–21 Hz) in the anterior cingulate cortex. In the same study, the depression level in tinnitus patients correlated with an elevated alpha (10.5–12.5 Hz) current source density in the left frontal cortex. Another brain structure (the para-hippocampal gyrus) was considered as a node connecting the areas involved in tinnitus-related distress and depression. As this study suggested, in tinnitus patients both these emotional states are processed by specific neural circuits which are part of a greater common network.

Further evidence for specific alterations in neural networks associated with bothersome tinnitus comes from resting-state fMRI studies in which functional connectivity (FC)–i.e. temporal correlations of low-frequency fluctuations in the BOLD response [69]–are calculated. High-distress tinnitus was found to be associated with negative FC between the sensory cortices (auditory and visual)–i.e. when the signal decreases in one of these areas, it increases in the other [51]. On the other hand, non-bothersome tinnitus is considered to not significantly affect the FC pattern [60]. Tinnitus-related distress is thought to be positively associated with increased FC between the posterior cingulate cortex and right medial prefrontal cortex belonging to the default mode network (DMN) [52] that is typically engaged at rest and suppressed during performance of a mental task [62], or with enhances connectivity of the superior temporal gyrus with the amygdala within one hemisphere (right or left), which suggests a role for

interaction between the auditory and limbic networks in emotional reaction to tinnitus [53]. There is also evidence of a positive correlation between tinnitus-related distress and FC of the right superior temporal gyrus and the cerebellum, which might reflect a dysfunctional filtering of unpleasant sounds [55]. In a recent study on a large tinnitus patient group ($n$ = 135), highly bothersome tinnitus was associated with increased FC within the right executive control network encompassing the frontal and parietal regions, as well as greater connectivity between this network and the left executive control, default mode, auditory, and salience networks [57]. Since the executive control network is considered to be involved in various attentional processes (e.g. allocating top–down attentional resources, cognitive control, externally oriented awareness [70,71], the authors suggested that high-distress tinnitus could be related to attention to the sound.

Brain activity of high- and low-distress tinnitus patients has also been investigated during fMRI cognitive/emotional tasks. Patients with high distress (compared to low-distress) tinnitus showed increased activation in the left mid frontal gyrus in response to tinnitus-related sounds *versus* neutral sentences [72], or in the right insula and orbitofrontal cortex in the tinnitus-related sounds *versus* neutral words contrast [73]. Decreased activation of the frontal areas in patients with more bothersome tinnitus, relative to those who do not suffer from it so much, was found when they listening to affective sounds not associated with tinnitus [74]. Overall, these studies suggest a role for the frontal cortex and insula in the processing of emotional auditory stimuli in patients with troublesome tinnitus.

In the present study the EEG power and current density from signal recorded when focusing on tinnitus was contrasted with the signal acquired when just concentrating on other internal sensations from one's own body. Such comparisons should allow one to identify the neural correlates of emotional reaction to tinnitus and the amount of attentional resources allocated to it. The literature in this area is scarce. The findings to date are that there is a lack of significant difference in global spectral power or source of the EEG signal when a comparison is made between active listening to tinnitus and the resting state [75]. We expect that the most pronounced differences between the above conditions to be found in the alpha band over the auditory areas in patients with high tinnitus-related distress. The rationale for this statement comes from a study suggesting that augmented alpha rhythm in sensory cortices reflects inhibition of sensory information processing in these regions [76]. Since in the present study the participants are supposed to be concentrated and attentive, increased alpha activity might be also observed over the areas that form the networks subserving attention, mainly, the prefrontal and parietal cortices [77]. In the face of evidence for reduced alpha power in response to emotional stimuli [78] we also assume that concentrating on bothersome tinnitus, i.e. an unpleasant and/or annoying sound, might be associated with more active information processing and, hence, lower alpha activity, compared to the focusing on one's own body (a neutral stimulus). The above effect might be, therefore, seen over the regions involved in a relaxation and concentration on inner world [62] and/or unpleasant sensory experience [79,80].

In the current study it is also hypothesized that in comparison to focusing on something without any special meaning (e.g. one's own body), the state of concentrating on bothersome tinnitus produces higher beta activity. Increased high-frequency oscillations have been found in response to salient or unpleasant stimuli [81,82]. Moreover, beta rhythm is thought to be associated with stress and anxiety [83] as well as concentration and arousal [84,85]. Therefore, high-distress tinnitus patients might be more occupied by their own tinnitus sound which contributes to increased beta activity over the brain representations regions of attentional and/or emotional processing.

## Materials and methods

### Participants

Eighty-seven participants with chronic ($\geq$ 6 months) subjective tinnitus were recruited from patients of the Institute of Physiology and Pathology of Hearing (IPPH), Warsaw/Kajetany, Poland. Twenty of them were excluded due to excessive motion or muscle artifacts in the EEG signal and/or missing data in the questionnaires assessing tinnitus distress, depression, or anxiety. The remaining 67 patients (36 women and 31 men, mean age = 50.3 ± 12.9 years, age 17.7–70.6 years) did not suffer from any serious illness (hyperacusis, Meniere's disease, neurological/psychiatric disorder, or major medical disease such as cancer or thyroid dysfunction), had no history of head trauma, seizure, stroke, smoking, or alcoholism and had not taken any medications affecting the central nervous system in the past 6 months. All subjects had not used any form of meditation (e.g. mindfulness, yoga) prior to participation in the study.

Each subject gave written informed consent to take part in the study after all procedures had been fully explained. The study was approved by the Ethics Committee of the IPPH and was congruent with the principles of the Declaration of Helsinki.

Participants were split into two groups according to the median value obtained in the Polish version of the Tinnitus Handicap Inventory (THI) [86]. As a result, there was a high tinnitus-related distress group (HD = 34) and a low tinnitus-related distress group (LD = 33). The detailed characteristics of LD and HD patients are presented in Table 1.

### Medical interview and audiological examination

All procedures were carried out at IPPH. Each patient participated in a medical interview followed by otolaryngological examination conducted by an ENT specialist. Audiometric thresholds were measured for 0.125–8 kHz in accordance with standard procedures provided by the British Society of Audiology [87]. Average thresholds for 0.125, 0.25, 0.5, 0.75, 1, 1.5, 2, 3, 4, 6,

**Table 1. Statistical comparison of patients with low tinnitus-related distress (LD) and high tinnitus-related distress (HD).**

|  | LD (n = 33) | HD (n = 34) | t, z, or $\chi^2$ value |
|---|---|---|---|
| **Tinnitus characteristics** |  |  |  |
| THI (M ± SD) | 27.37 ± 7.41 | 58.26 ± 14.6 | 7.04*** |
| Tinnitus duration (months, M ± SD) | 57.52 ± 55.78 | 66.35 ± 64.1 | .736 |
| Tinnitus location (left/right/bilateral/head) | 6/5/19/3 | 4/3/22/5 | 1.61 |
| **Demographic data** |  |  |  |
| Age (years, M ± SD) | 51.94 ± 11.77 | 47.97 ± 13.98 | .99 |
| Sex (women/men) | 19/14 | 17/17 | .387 |
| **Depression and anxiety** |  |  |  |
| BDI-II (M ± SD) | 9.03 ± 8.33 | 13.29 ± 8.02 | 2.59* |
| STAI X-1 (M ± SD) | 37.61 ± 8.37 | 42.18 ± 10.75 | 1.81 |
| STAI X-2 (M ± SD) | 41.58 ± 8.11 | 48.03 ± 8.73 | 3.16** |
| **Hearing loss (dB HL)#** |  |  |  |
| Right ear | 17.41 ± 13.07 | 16.51 ± 10.12 | .063 |
| Left ear | 17.41 ± 10.65 | 17.56 ± 10.59 | .213 |

* $p < 0.05$

** $p < 0.01$

*** $p < 0.001$

THI—Tinnitus Handicap Inventory; BDI—Beck Depression Inventory; STAI–State Trait Anxiety Inventory.

# Mean values of hearing loss both for the right and left ear were calculated for 0.125, 0.25, 0.5, 0.75, 1, 1.5, 2, 3, 4, 6, and 8 kHz

and 8 kHz were calculated separately for the right and left ear. The outcomes of the pure tone audiometry in LD and HD groups are presented in Fig 1.

## Questionnaires

Patients were requested to fill in the THI [88], Beck Depression Inventory (BDI-II) [89], and State Trait Anxiety Inventory (STAI) [90] (the order of administration was randomized across participants). Tinnitus distress was determined by means of a Polish version of THI, a reliable tool widely used to assess how tinnitus affects daily living. THI consists of 25 questions concerning limitations of social/occupational, physical, and emotional functioning as well as catastrophic reactions to the presence of tinnitus. A *yes* response to a question is awarded 4 points, *sometimes* 2 points, and *no* 0 points, with higher scores representing greater tinnitus distress. Subjective depression was assessed using BDI-II which is a 21-item tool asking for responses on a 3-point scale, with higher scores reflecting higher intensity of symptoms. STAI includes two 20-item scales, one of which (X-1) is referred to anxiety as a state and the other (X-2) measures anxiety as a personal trait. Each patient responded on a 4-point scale (scores 1–4) where higher scores indicate higher level of anxiety.

## EEG data acquisition and pre-processing

The EEG signal was recorded in an acoustic-shielded room. Each participant was seated in a comfortable chair in an upright relaxed position. Prior to data recording, he or she was instructed to refrain from any movements (especially those involving the face, e.g. eyes, forehead, neck, and jaw). The EEG session consisted of two 3–4 min blocks during which patients were asked to focus attention (while keeping the eyes closed) and analyze sensations associated either with: 1) tinnitus sound (the tinnitus-focus condition, TFC) or 2) their own feet (the body-focus condition, BFC). The order of these two blocks was counterbalanced across the participants. The TFC and BFC in each patient were separated by a short (ca. 1-min) break, mainly conversation with the experimenter.

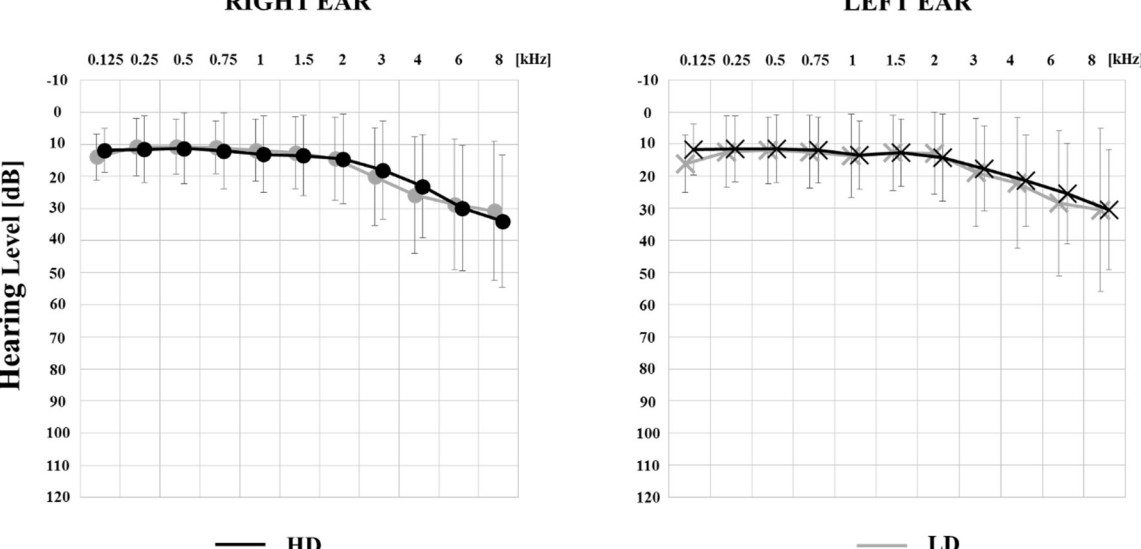

**Fig 1. Audiograms: Average left and right ear hearing thresholds in the high tinnitus-related (HD) and low tinnitus-related (LD) distress groups.** Errors bars show standard deviations of the individual means.

The EEG data were acquired from 19 channels (ElectroCap International, Inc., Eaton, USA), placed according to the 10–20 system [91] with the use of a 21-channel apparatus (Mitsar, St. Petersburg, Russia). The signal was referenced to linked earlobes and Fpz electrode was the ground electrode. The impedance was kept below 10 kΩ during the whole data registration. Sampling rate was 250 Hz and the signal was digitally filtered in the 0.3–70 Hz band.

The EEG signal from each condition was pre-processed using Neuroguide 2.9.4 software (Applied Neuroscience, St. Petersburg, FL). The data pre-processing steps were as follows: 1) the signal was filtered between 0.5 and 50 Hz; 2) artifacts were removed with the use of a semi-automatic algorithm implemented in the program (in each patient a manually selected 10-sec artifact-free epoch of the whole acquired EEG signal served as a template for automatic selection of segments without artifacts) [92]; 3) statistical analysis of the selected EEG segments using the split-half and test–retest reliability methods [93,94]. Only artifact-free EEG segments with a split-half reliability greater than 95% and test–retest reliability greater than 90% as well as a length $\geq$ 1 min were accepted and included to further analysis [92].

All data used in this study is available on figShare database with the following DOI: 10.6084/m9.figshare.12643403.v1.

## QEEG data analysis

QEEG data analysis was also performed with the use of the Neuroguide 2.9.4 software. First, a fast Fourier transform (FFT) was performed on the EEG signals to determine the absolute power in the following bands: delta (1–4 Hz), theta (4–8 Hz), low alpha (8–10 Hz), high alpha (10–12 Hz), as well as low (12–15 Hz), middle (15–18 Hz), and high (18–25 Hz) beta and gamma (30–40 Hz), for both the TFC and BFC. A 75% sliding window was used to compute the FFT in consecutive 2-s epochs (i.e. 256 points were overlapped in 500-ms steps (64 points) in order to minimize the effects of FFT windowing [95]). The EEG absolute power in each frequency band was averaged separately for 7 clusters (Fig 2).

The EEG data were checked for departures from normality with the use of a Shapiro–Wilk test. Since most distributions showed significant abnormality (basically due to right skewness), our data were normalized by means of square, cube square, or Lg10 transforms prior to statistical analysis.

A repeated measures ANOVA with the "frequency band" (delta, theta, low and high alpha, low, middle, high beta and gamma), "condition" (TFC and BFC), and "cluster" (LA, RA, LM, RM, CE, LP, RP) as repeated factors, as well as the "tinnitus" (LD and HD) as a between-subject factor was performed and only $p$ values based on permutation methods were reported. Using the formula notation in R, the following model: POWER ~ TINNITUS*FREQUENCY BAND*CONDITION * CLUSTER + Error (SUB/(FREQUENCY BAND*CONDITION* CLUSTER)) + TINNITUS was fitted using *aovperm*() function implemented in R/*permuco* package [96] (100000 permutations). Post-hoc pairwise *t*-tests calculated with the use of R/ *emmeans* package [97] were bootstrapped (Boot() function implemented in R/*car* package, 100000 bootstrap samples) in order to deconstruct significant interaction effects, and the $p$-values from these bootstrap comparisons were adjusted using the false discovery rate (FDR) approach [98].

## sLORETA analysis

Standardized low-resolution brain electromagnetic tomography (sLORETA) was applied to determine the intracerebral sources of bioelectrical brain activity recorded at the scalp. To prepare EEG signal for sLORETA analysis the Neuroguide ver. 2.9.4 software was used. Pre-selected artifact-free segments of EEG signals in each frequency band were exported to special

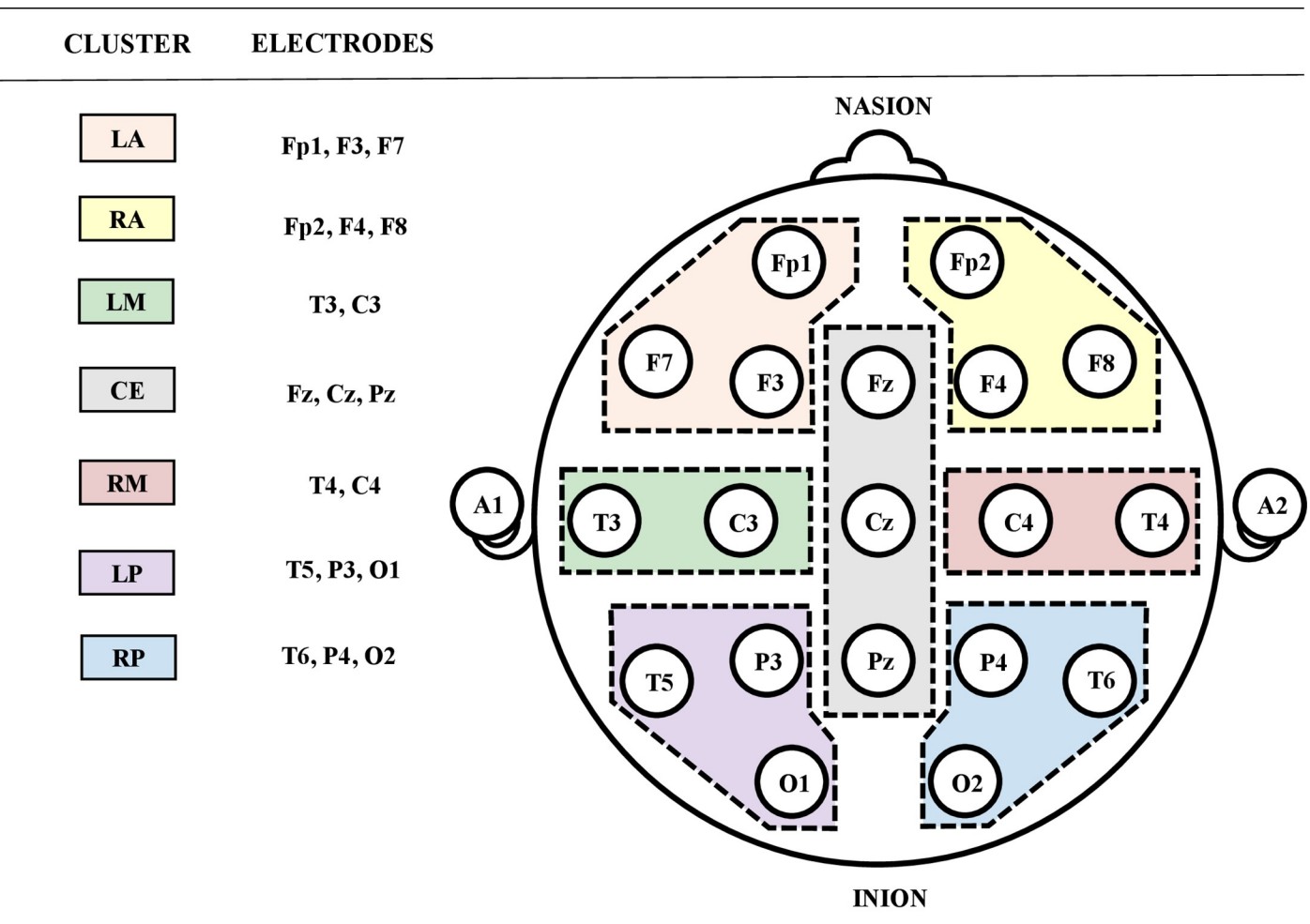

**Fig 2. A scheme of electrode montage.** Clusters were presented from anterior (left, cluster LA orange and right, cluster RA yellow), left middle (cluster LM green), central (cluster CE gray), right middle (cluster RM red) and posterior (left, cluster LP purple and right, cluster RP blue).

ASCI files containing successive 256 points time series of signals recorded in all 19 channels. Overlapping windows method, implemented in the Neuroguide, was applied during export time series to minimize the FFT windowing effects by overlapping 256 point × 19 channel EEG segments by 75% in ASCII format [99]. Next, the time to frequency domain conversion was applied to pre-prepared ASCI files. The cross-spectral files for every frequency band were determined for 19 electrodes, 256 time points and the sampling rate of 128 Hz with the use of the EEG cross-spectral maker implemented in sLORETA utilities. Finally, the cross-spectra prepared in previous step, were transformed into sLORETA files containing information about the current density vector field power specific for each subject.

A standardization used by sLORETA algorithm allows to solve the inverse solution with zero localization error [100] and the point EEG sources can be find without any previous assumptions as to the numbers of modelled sources and their localizations [100,101]. As a result of sLORETA the distribution of current density (μA/mm2) is determined within solution space formed by 6239 voxels (voxel size: 5 × 5 × 5 mm) and restricted to cortical gray matter, hippocampus and amygdala, defined by the digitized Montreal Neurological Institute

(MNI) brain template [102]. The sLORETA algorithm solves the inverse problem by assuming related orientations and strengths of neighboring neuronal sources (represented by adjacent voxels). The method has been proved as an efficient functional brain activity mapping tool with a high temporal resolution and relatively high spatial resolution [100]. The localization properties of sLORETA have been previously validated (e.g. [103,104]).

In our study the current density distributions between the conditions (TFC *versus* BFC) and groups (HD *versus* LD), were compared with the use of non-parametric permutation tests (SnPM) of sLORETA. The computations were performed on a whole brain by sLORETA statistical contrast maps using multiple voxel-by-voxel comparisons method with 5000 permutations. As a result, the maps of log *t*-ratio statistics for each frequency band were generated. The statistical thresholds, corrected for multiple testing, were applied ($p < 0.05$ and $p < 0.1$, 2-tailed) [105] and the voxels with *t*-values that exceeded the pre-defined statistical threshold were used to identify the Brodmann areas (BA) where brain activity was significantly different between conditions or groups.

## Results

### QEEG analysis

The ANOVA results are shown in Table 2. We found a substantial 3-way interaction effect: "frequency band" × "condition" × "cluster" ($F_{42, 2730} = 1.54$, $p = 0.0141$) and a 4-way interaction: "frequency band" × "condition" × "cluster" × "tinnitus distress" ($F_{42, 2730} = 1.71$, $p = 0.0037$). The results of *post-hoc* analysis were presented in S1–S6 Tables.

**Tinnitus *versus* body focus condition.** In the whole study sample delta power in the LA cluster was significantly ($p = 0.02$, *FDR*-adjusted) higher in the TFC (M = 0.972, SE = 0.072

**Table 2. The results of repeated-measures ANOVA.** P values based on permutation tests for repeated measures ANOVA.

| | df$_{Num}$ | df$_{Den}$ | *F* | permutation *p* |
|---|---|---|---|---|
| Tinnitus distress | 1 | 65 | 2.34 | 0.1284 |
| Frequency | 7 | 455 | 94.75 | 0.0001*** |
| Frequency × tinnitus distress | 7 | 455 | 1.22 | 0.2924 |
| Condition | 1 | 65 | 0.92 | 0.3364 |
| Condition × tinnitus distress | 1 | 65 | 0.58 | 0.4499 |
| Cluster | 6 | 390 | 50.62 | 0.0001*** |
| Cluster × tinnitus distress | 6 | 390 | 2.14 | 0.0494* |
| Frequency × condition | 7 | 455 | 1.27 | 0.2642 |
| Frequency × condition × tinnitus distress | 7 | 455 | 0.44 | 0.8848 |
| Frequency × cluster | 42 | 2730 | 46.74 | 0.0001*** |
| Frequency × cluster × tinnitus distress | 42 | 2730 | 0.35 | 1.0000 |
| Condition × cluster | 6 | 390 | 0.96 | 0.4521 |
| Condition × cluster × tinnitus distress | 6 | 390 | 0.81 | 0.5608 |
| Frequency ×condition × cluster | 42 | 2730 | 1.54 | 0.0141* |
| Frequency ×condition × cluster × tinnitus distress | 42 | 2730 | 1.71 | 0.0037** |

df$_{Num}$ indicates degrees of freedom numerator, df$_{Den}$ indicates degrees of freedom denominator, p values were computed for permutations in repeated-measures ANOVA.

*p < 0.05

**p < 0.01

***p < 0.001.

µV) than in the BFC (M = 0.943, SE = 0.020 µV). Furthermore, a power in the gamma frequency band over the RA region in the TFC was increased relative to the BFC (*p* = 0.013, *FDR-corrected*, for TFC: M = 0.269, SE = 0.034 µV and for BFC: M = 0.223, SE = 0.033 µV).

The results of comparisons between the bioelectrical activity recorded in the TFC and BFC in the HD and LD groups are shown in Fig 3 and included in S3 and S4 Tables.

For the HD group in the TFC, compared to BFC, there was significantly (*p* = 0.034) decreased low alpha power over the LA (M = 0.686, SE = 0.068 µV for the TFC; and M = 0.734, SE = 0.065 µV for the BFC) and RA (*p* = 0.028, M = 0.709, SE = 0.07 µV for the TFC and M = 0.752, SE = 0.068 µV for the BFC) areas. These patients also demonstrated reduced high alpha band power over the RA (*p* = 0.05, for the TFC: M = 0.570, SE = 0.06 µV and for the BFC: M = 0.613, SE = 0.064 µV), CE (*p* = 0.026, for the TFC: M = 0.872, SE = 0.069 and for the BFC: M = 0.935, SE = 0.074), LP (*p* = 0.006, for the TFC: M = 1.001, SE = 0.082 µV and for the BFC: M = 1.07, SE = 0.086 µV) and RP (*p* = 0.003, for the TFC: M = 0.993, SE = 0.086 µV and for the BFC: M = 1.073, SE = 0.091 µV) regions. The HD patients in the TFC, relative to the BFC, produced enhanced gamma power in the RA cluster (p = 0.020, for the TFC: M = 0.245, SE = 0.048 and for the BFC: M = 0.198, SE = 0.047).

In the LD group only one significant effect was observed, namely, increased delta power over the LA area in the TFC compared to the BFC (p = 0.0320, for the TFC: M = 0.976, SE = 0.031 and for the BFC: M = 0.954, SE = 0.026).

**High *versus* low tinnitus distress.** The differences in frequency band power between the HD and LD groups in the BFC and TFC are shown in Fig 4 and S5 and S6 Tables.

During the BFC in the LM area low beta power was significantly (*p* = 0.026) higher in the HD group (M = 0.538, SE = 0.045 µV) than in the LD group (M = 0.395, SE = 0.046 µV). In the TFC, there was also increased (*p* = 0.019) low beta power in the same region in the HD (M = 0.539, SE = 0.044 µV) *versus* LD (M = 0.392, SE = 0.044 µV) group. Finally, only in the BFC enhanced power in the low-frequency beta band (*p* = 0.045) was found in the LP cluster for the HD group (M = 0.707, SE = 0.055 µV), relative to the LD patients (M = 0.549, SE = 0.056 µV).

In the BFC middle beta power over the CE and LP regions was significantly increased in the HD (for the CE: *p* = 0.048, M = 0.501, SE = 0.046 µV and for the LP: *p* = 0.022, M = 0.484, SE = 0.047 µV) compared to the LD group (for the CE: M = 0.371, SE = 0.047 µV and for the LP: M = 0.329, SE = 0.048 µV). In the TFC, significantly (*p* = 0.025) higher middle beta power in the HD (M = 0.476, SE = 0.047 µV) compared to the LD patients (M = 0.326, SE = 0.047 µV) was observed only in the LP area.

The HD group in the BFC showed increased high beta power compared to the LD patients in the CE area (*p* = 0.029, M = 0.691, SE = 0.038 µV for HD patients and M = 0.576, SE = 0.039 µV for LD patients) and the LP area (*p* = 0.02, for HD: M = 0.656, SE = 0.039 µV and for LD: M = 0.528, SE = 0.04 µV), whereas in the TFC this effect was only found to be significant (*p* = 0.005) in the LP region (for HD: M = 0.647, SE = 0.037 µV and for LD: M = 0.5, SE = 0.038 µV).

## sLORETA

Only in the HD group sLORETA analysis showed significant between-condition differences for low alpha (8–10 Hz) and middle beta (15–18 Hz) frequency bands (Fig 5, Table 3). Specifically, in these patients during the TFC, compared to BFC, there was a decrease of current density in the low alpha band in the right frontal area (medial, superior, middle and inferior frontal gyrus) and in the right anterior cingulate (threshold log *t*-statistics = 4.150 for *p* < 0.05; and *t* = 3.889 for *p* < 0.1). Conversely, middle beta current density in the HD group in the

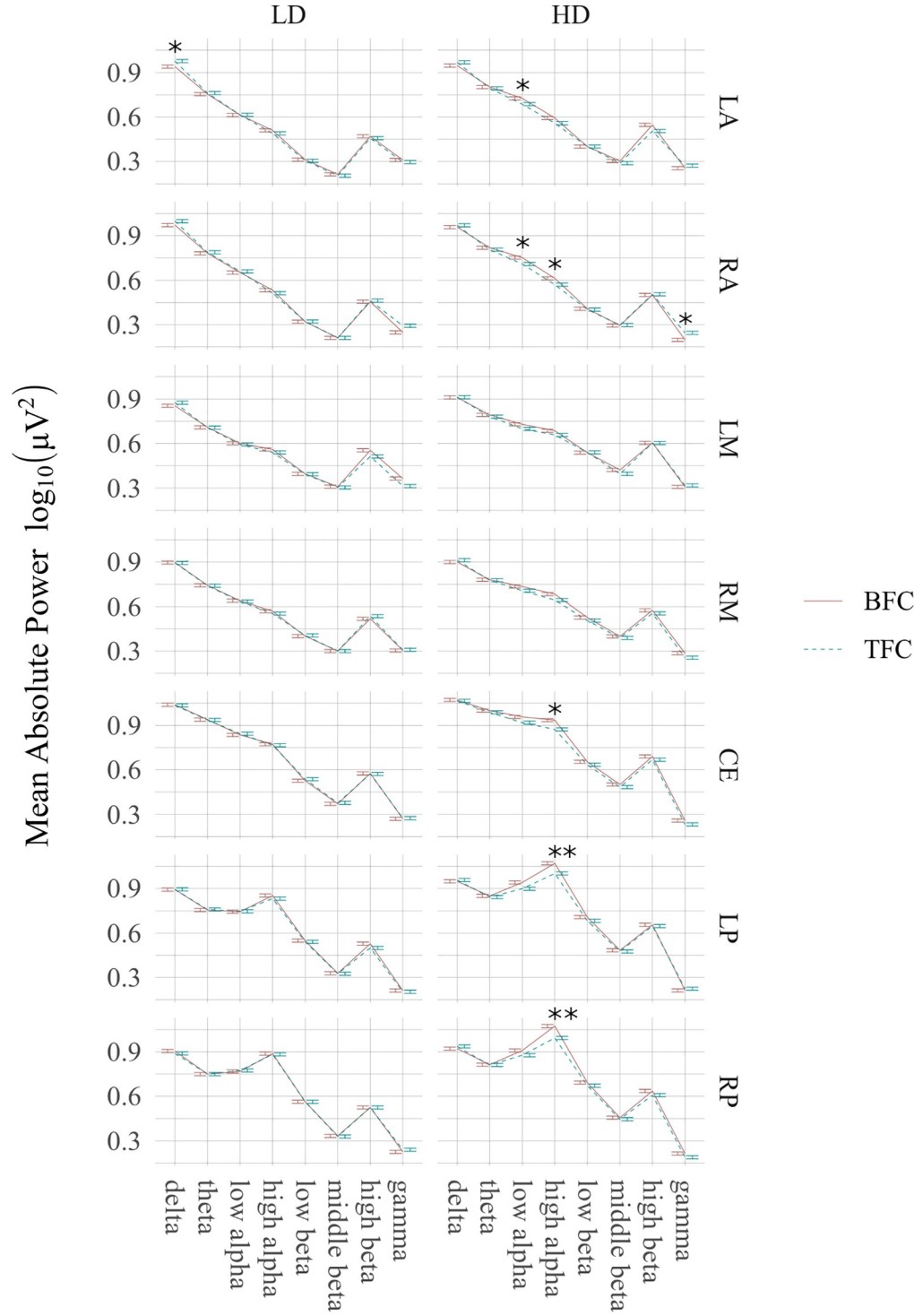

**Fig 3. Comparisons of particular frequency band (delta, theta, low and high alpha, low, middle and high beta, gamma) power between experimental conditions (tinnitus focus condition, TFC, and body focus condition, BFC) in selected clusters (LA–left anterior, RA—right anterior, LM—left middle, RM—right middle, CE–central, LP— left posterior, RP–right posterior).** The significant differences are marked with "stars" (* p < 0.05, ** p < 0.01).

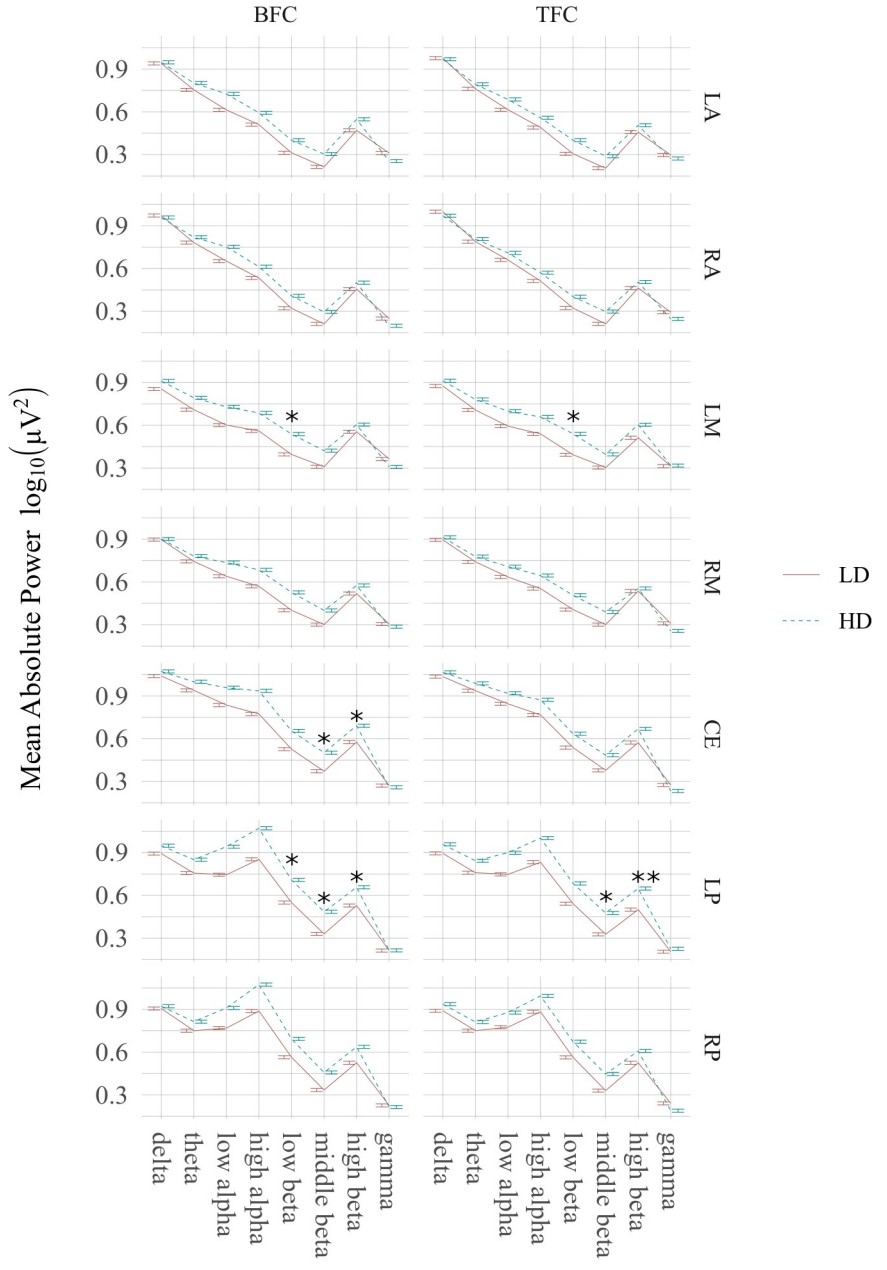

**Fig 4. Comparisons of particular frequency band power (key as in previous figure) between patients with high (HD) and low (LD) tinnitus-related distress in selected clusters (key as in previous figure).** The significant differences are marked with "stars" (* p < 0.05, ** p < 0.01).

TFC, relative to the BFC, was increased in bilateral posterior cingulate and left precuneus (Fig 5, Table 3) (threshold log *t*-statistics = 3.889 for *p* < 0.1). sLORETA analysis did not reveal any significant between-group (HD *versus* LD) differences.

A

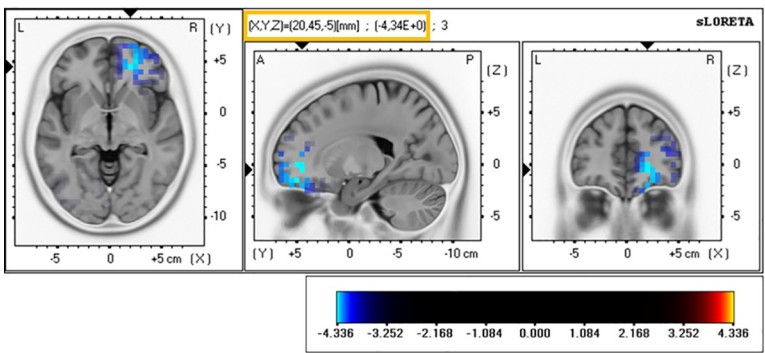

B

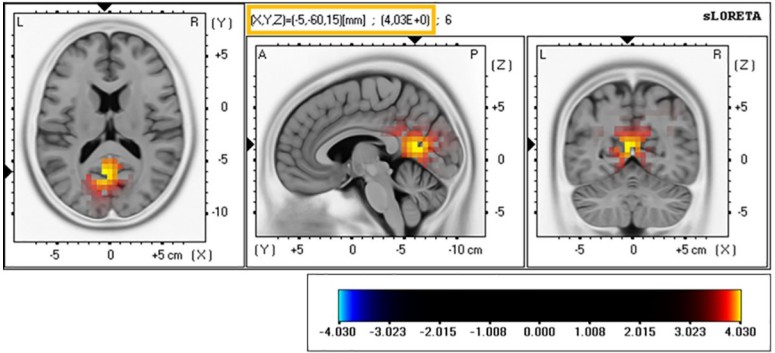

**Fig 5.** Significant results for current density amplitude analysis in the low alpha (8–10 Hz) band (A) and middle beta (15–18 Hz) band (B). In the high distress tinnitus patients sLORETA current source density in the low alpha band was lower in the tinnitus focus condition compared to the body focus condition in the right frontal/cingulate cortex and, in the same contrast, the current density in the middle beta (15–18 Hz) was higher in the posterior cingulate/precuneus area. T-statistic that are positive are displayed in red (the mean current density in the body focus condition is greater than the mean in the tinnitus focus condition) and those that are negative in blue (the mean current density in the body focus condition is lower than the mean in the tinnitus focus condition). The horizontal (left) sagittal (middle), and coronal (right) sections through the voxel with maximal t-statistic are presented. The maximal *t*-value and MNI coordinates of the voxel with this value are shown in the yellow frame above a sagittal section. The image shows significant outcomes only.

## Discussion

To our knowledge this is the first study demonstrating that active processing of bothersome tinnitus sensations coexist with alterations in the alpha (seen both in EEG power and current density analyses), middle beta and gamma rhythms (Figs 3, 5A, Table 3, S3 and S4 Tables). Furthermore, we found that patients with high and low tinnitus-related distress were different in terms of beta band power regardless of the experimental condition, i.e. while patients concentrating on their tinnitus and on sensations in another part of their body (Fig 4, S5 and S6 Tables).

### Focusing on highly distressing tinnitus alters the alpha (8–12 Hz), but also beta (15–18 Hz) and gamma (30–40 Hz) rhythms

Congruent with our expectations, differences between the EEG signal recorded in each condition were most pronounced in the patients with high tinnitus-related distress in the alpha and

**Table 3. Brain structures with significant differences in current density (µA/mm2) between the tinnitus focus condition and body focus condition in patients with high distress tinnitus.** The regions were identified by the voxel-by-voxel sLORETA non-parametric permutation test. The table includes the Brodmann's areas (BA) and brain regions, (Montreal Neurological Institute) MNI coordinates of voxels with the highest t-values and total number of activated voxels (*p*-values are marked with "stars").

| Band | BA | Region | Side | MNI coordinates | | | Max t-value | Total number of voxels |
|---|---|---|---|---|---|---|---|---|
| | | | | X | Y | Z | | |
| **Low Alpha** | 10 | Medial Frontal Gyrus | Right | 20 | 45 | -5 | -4.34** | 5 |
| | 10,11 | Superior Frontal Gyrus | Right | 20 | 50 | 0 | -4.34** | 40 |
| | 10,11 | Middle Frontal Gyrus | Right | 25 | 50 | -10 | -4,27** | 19 |
| | 32 | Anterior Cingulate | Right | 15 | 45 | -5 | -4.14* | 4 |
| | 11 | Inferior Frontal Gyrus | Right | 20 | 40 | -20 | -3.94* | 1 |
| **Middle Beta** | 23,29,30,31 | Posterior Cingulate | L, L/R, R | -5 | -60 | 15 | 4.03* | 16 |
| | 23,31 | Precuneus | L,L/R | 0 | -60 | 20 | 3,89* | 3 |

p < 0.05**

p < 0.1*

beta frequency bands. Specifically, only in this group, when the TFC was compared to the BFC, there was a significant decrease of the low-frequency alpha (8–10 Hz) power over the anterior area, and reduced activity within the high alpha band (10–12 Hz) over right anterior, left and right posterior and central regions (Fig 3, S3 and S4 Tables). In the TFC, relative to the BFC, the HD patients also showed increased gamma power but only at the electrodes positioned over the right frontal cortex. Additionally, for the above contrast, sLORETA analysis revealed reduced low alpha current density amplitude in right frontal area and right anterior cingulate and enhanced density in the middle beta band in the precuneus (Fig 5, Table 3).

The alpha rhythm (8–12 Hz) is thought to reflect the balance between excitatory and inhibitory neuronal processes: reduced activity in this frequency band is associated with excitation or more active cognitive processing whereas enhanced activity is associated with inhibition (the alpha-inhibition hypothesis, [74,75]). By interpreting our outcomes according to this hypothesis we claim that reduced alpha power under the TFC compared to the BFC in the HD patients, may reflect higher cortical excitability and/or a more detailed analysis of tinnitus relative to the sensations derived from a part of one's body. The strength of this explanation is enhanced when we take note that a relaxed state of mind is more easily accomplished when, as in the present study, subjects are asked to keep their eyes closed. In addition, focusing on one's own body is an important part of mindfulness therapy [106] and the presented outcomes, showing enhanced alpha power over the anterior and posterior cortices when focusing on one's own body, are consistent with the evidence on augmented alpha activity in the frontal and/or parieto-temporo-occipital cortex during various meditation states [107–112].

In our study reduced high alpha power (10–12 Hz) over the central and posterior areas in the HD patients, when the TFC was compared to the BFC, might reflect greater semantic processing demands in the TFC conditions [113]. In other words, decreased high alpha power in the TFC might indicate that tinnitus was perceived as a meaningful sound. This interpretation of our findings is in agreement with previous evidence of a strong decrease in alpha activity (8–12 Hz) in the auditory cortex when a salient sound is heard [114]. Furthermore, when we take into account that affective stimuli, in comparison to neutral ones, induce more reduced posterior alpha rhythm [115], diminished power within this frequency band in the TFC in our study might also reflect an emotional reaction to tinnitus.

The analogy between phantom sound (tinnitus) and phantom pain are both thought to be due to thalamo-cortical dysrhythmia [30], and our results are in accordance with the evidence

on reduced alpha and enhanced gamma activity when attention is devoted to the sensation of pain [116]. Decreased alpha might reflect the degree to which the thalamo-cortical "gate" is opened and therefore how relevant information reaches the cortex and how it is then processed. In our study, concentrating on bothersome tinnitus keeps the gate open, resulting in weak inhibition of tinnitus-related information that might manifest as decreased power and/ or current density within the alpha band accompanied by increased gamma activity.

The similarities between tinnitus and chronic pain (i.e. both are variable subjective sensations caused by peripheral lesions, the patients also share a psychological profile characterized by increased tendency to depression and anxiety [41]) implicate a common neural foundation resulting from the presence of chronic and unpleasant symptoms. At first glance, in the case of alpha rhythm, our results do not appear consistent with either previous evidence on troublesome tinnitus [56,59,65] or the literature on chronic pain (see: [117] for a review) which both suggest elevated rather than reduced activity within this band at rest. Chronic high tinnitus-related distress is supposed to be apparent during the resting state due to the absence of any stimulus presentation and the fact that it is hard to keep attention away from persistent and annoying sound. Therefore, the high distress tinnitus patients when concentrating on tinnitus are supposed to produce the EEG activity comparable to that observed at "pure" rest.

The resting state might be, however, considered just as a state of relaxed, calm mind. In this case it could be more similar to the condition, explored in the present study, when patients are requested to focus on the own body (a legitimate therapeutic technique to achieve a state of relaxed, calm mind). When we take into account the resembles between both these states, our results in the alpha frequency band (increased activity under the BFC relative to the TFC) are congruent with previous findings [56,59,65].

In our study decreased current density within low alpha band in the HD patients was observed under the TFC, compared to the BFC, in the right frontal and anterior cingulate cortices (Fig 5A, Table 3). Since these regions are thought to form the pain network [118], the above-described differences might not be unique for tinnitus but rather associated with chronic, troublesome ailment. Additionally, our results are consistent with the valance hypothesis suggesting the right hemisphere dominance for negative emotions and left hemisphere dominance for positive emotions [119]. The above differences may, then, reflect affective aspects of tinnitus.

We found increased gamma (30–40 Hz) power over the right anterior area in the TFC relative to the BFC in all patients but especially in bothersome tinnitus sufferers (Fig 3, S3 and S4 Tables). Gamma activity associated with tinnitus-related distress has been relatively well documented [46,66,67,120], however, comparable number of evidence failed to demonstrate any relationship [56,59,65,121]. Since in our study in the HD group the between-condition difference in gamma power was found only in one cluster (and also the high and low distress tinnitus groups did not substantially different in the activity within this frequency band) it is hard to explain increased gamma, observed in the TFC, in terms of tinnitus-related distress. We rather claim that higher gamma, thought to be associated with a state of active information processing [122] may be indicative for greater cognitive/emotional involvement in the analysis of sensations associated with bothersome tinnitus. Notably, the gamma effect was also observed not only for the HD patients but in the whole study group which might indicate that more active processing of tinnitus rather than related distress, is reflected in this high-frequency range.

Similarly to the outcomes of the present study, in the literature on chronic pain enhanced gamma activity over the prefrontal cortex has been found at rest [123–125] or when noxious-attended condition was contrasted with the resting state [126]. Therefore, increased gamma power in the HD patients in the TFC, compared to the BFC, might reflect unpleasantness and

distress not specifically related to tinnitus. However, since the alterations in chronic pain have been mostly described for much higher gamma band (> 60 Hz) than the frequency range analyzed in our study, we cannot discuss our outcomes in terms of the analogy between tinnitus and chronic pain conditions.

Although we did not observe any significant differences between the TFC and BFC in beta power, in this contrast we found higher current source density in the middle beta frequency range (15–18 Hz) in the precuneus/posterior cingulate but only in the HD group (Fig 5B, Table 3). Since this rhythm, just as the precuneus activity, is thought to be involved in attentional processes [127,128], the above effect most probably reflect more cognitive resources allocated to tinnitus as compared to sensations derived from the own body.

As expected, in the present study there were not much significant differences between bioelectrical activity in the BFC and TFC in patients with low tinnitus-related distress. Thus, non-bothersome tinnitus appears not to be an affective, meaningful sound deserving of special attention. In this case it will not be processed differently to any other relatively neutral stimulus such as sensations from one's own body. However, we observed slightly enhanced delta power in the TFC compared to the BFC over left anterior area only in the LD patients (Fig 3, S3 and S4 Tables). Since frontal delta rhythm is thought to increase in various tasks (e.g. semantic processing, mental calculation, see: [129] for a review), the above effect might reflect more cognitive efforts devoted to active listening of tinnitus (definitely the forced concentrating on a tinnitus sound is not a natural situation for the patients who no longer pay attention to it) than focusing on one's body part.

## Beta rhythm (12–25 Hz) is increased in high *vs* low tinnitus-related distress

The high-distress compared to low-distress tinnitus patients showed increased low-frequency beta power (12–15 Hz) over the left-middle area in both experimental conditions whereas in the BFC this effect was observed only over the left posterior region (Fig 4; S5 and S6 Tables). Low-frequency beta power over the sensorimotor cortex is called the sensorimotor rhythm (SMR), which is thought to reflect an alert state of mind during active inhibition of motor behavior [130]. It becomes desynchronized while imagining body movement [131]. Neurofeedback trainings aimed at increasing SMR has been used to lower seizure activity [132], enhance semantic memory [133], and sustain attention [134,135]. In the present study, the HD compared to the LD group showed, under both conditions, increased low beta power at the C3 and T3 electrodes (i.e. over the sensory and motor cortices). It is therefore possible that the between-group differences observed in our study in the range of 12–15 Hz were from the SMR. In this case, the HD group may have put more effort into inhibiting body movement compared to the LD group (although prior to recording all subjects were asked to remain still to avoid EEG artifacts).

The 12–15 Hz rhythm outside the motor cortex is thought to be related to relaxed attention such as those involved in pleasant activities [136]. In the present study, there was greater low beta power over the left temporo-parieto-occipital cortex in the high-distress tinnitus patients compared to the low-distress patients, but only when they were concentrating on sensations from their own body. Therefore, these between-group differences might just indicate that not focusing on tinnitus is a much more pleasant state for the HD group than the LD group. This conclusion follows naturally from considering that patients with more bothersome tinnitus could feel relief when they don't have to monitor and analyze a distressing sound in their head. Finally, since the posterior brain area is thought to be involved in various attentional processes [137], enhanced low beta power in this region in the HD, compared to the LD patients, might also indicate greater attentional resources allocated to the BFC's task in the HD group.

In the present study the middle (15–18 Hz) beta power in both BFC and TFC was increased over the left posterior area in the HD compared to the LD group (Fig 4, S5 and S6 Tables). This

frequency band is thought to reflect activity of the noradrenergic system, which is involved in sustained attention and working memory [128]. Previous studies have shown that an enhancement of the 15–18 Hz rhythm after neurofeedback training correlated with better continuous performance [135] or sustained attention [134,138,139]. Augmented beta power was also accompanied by higher P300 amplitude in an oddball paradigm [134,135], which might also be considered an index of higher cortical excitation [140] or better working memory [141]. In this context, increased middle beta power in the HD, relative to the LD patients, in the BFC and TFC, might reflect more attentional resources engaged in these conditions by the first of these groups.

In the present study augmented low and middle beta power in bothersome tinnitus (compared to non-bothersome tinnitus) was observed in the left hemisphere, especially in the posterior area. These outcomes might be explained in terms of left hemispheric dominance in selective attention to fine-grained auditory information [142]. Accordingly, patients with high-distress tinnitus, especially when they are asked to focus on it, might be able to notice even very subtle characteristic in its sound, which mainly involves the left auditory cortex. The advantage of the left posterior region observed in the BFC might, on the other hand, result from more careful and detailed analysis of sensations derived from own body by the HD group.

Relative to LD patients, our HD group demonstrated enhanced high beta (18–25 Hz) over the left posterior area in both BFC and TFC, but only over the central region under the BFC. High-frequency oscillations are thought to reflect stress, anxiety, and increased cortical hyperactivity [83,143]. The locally augmented amplitude of the high beta band is also related to extensive muscle tension, and is most apparent in temporal brain regions [144]. Therefore, the increased high beta rhythm in our patients with bothersome tinnitus might be evidence of anxiety or increased cortical arousal rather than of tinnitus perception itself.

Our finding that the HD group had, under both experimental conditions, enhanced high beta activity over the left, but not the right, posterior area is consistent with evidence for involvement of the left temporal region in auditory hallucinations [145]. Tinnitus can be considered an hallucination, since it is present in the absence of any actual sound. Musical phenomena in tinnitus patients are thought to be due not to psychosis but to their emotional experience [146]. Hence, increased higher beta power over the left posterior area in our HD group, compared to the LD group, might indicate that high-distress tinnitus, relative to a non-bothersome phantom sound, is a clear and strong auditory phenomenon.

## Limitations of the study

Including to the study only the patients with tinnitus (without healthy controls) deprived us of the ability to draw reliable inferences about "increased" or "decreased" power within particular frequency bands in tinnitus. Specifically, without controls we could only conclude that bioelectrical activity was lower or higher in one clinical group relative to the other, but of course it is possible that in both these groups the compared parameter (e.g. band power) is abnormal. Thus, in future studies it is recommended that healthy controls be included whose only task is to concentrate on the body part.

When viewing the results one should be also acknowledged that any EEG based approach does not allow to analyze subcortical activity which constrains the effects to cortical sources.

## Conclusions

The most pronounced differences between concentrating on bothersome tinnitus and focusing on a one's body part were found in the high-distress tinnitus patients, in the alpha frequency band (i.e. decreased frontally distributed low alpha power and reduced high alpha rhythm over the central and posterior areas). These effects might reflect higher cortical excitability or more

active cognitive and emotional processing of tinnitus. Altered low alpha current density in the frontal and anterior cingulate cortices in the high tinnitus distress patients suggests sharing the attentional/emotional network with pain experience. Increased gamma power might reflect more active processing of bothersome tinnitus sensation whereas higher middle beta current density in the precuneus, observed in the tinnitus-focused condition compared to concentrating on the body, could be indicative for more attention resources allocated to troublesome tinnitus. Increased low beta (12–15 Hz) over the sensorimotor cortex and posterior middle and high beta (15–25 Hz) activity in the patients with high tinnitus-related distress, irrespectively of the condition, may reflect more thorough cognitive/emotional processing of sensations derived from the own body (not specifically associated with tinnitus).

## Supporting information

**S1 Table. The mean absolute power and standard error (in brackets) values for different frequency bands in the contrast: Body Focus Condition (BFC)** *versus* **tinnitus focus condition (TFC), calculated in each cluster for the whole study sample (n = 67).**
(PDF)

**S2 Table.** *Post-hoc* **pairwise t-tests results for the contrast: Body Focus Condition (BFC)** *versus* **Tinnitus Focus Condition (TFC) calculated in each cluster for the whole study sample (n = 67).** Items in bold are significant based on *p*-values.
(PDF)

**S3 Table. The mean absolute power and standard error (in brackets) values for different frequency bands in the contrast: Body Focus Condition (BFC)** *versus* **Tinnitus Focus Condition (TFC), calculated in each cluster separately for the low tinnitus-related distress (LD) and high tinnitus-related distress (HD) group.**
(PDF)

**S4 Table.** *Post-hoc* **pairwise *t*-tests results for the contrast: Body Focus Condition (BFC)** *versus* **Tinnitus Focus Condition (TFC), calculated in each cluster separately for the high tinnitus-related distress (HD) and low tinnitus-related distress (LD) group.** Items in bold are significant based on *p*-values.
(PDF)

**S5 Table. The mean absolute power and standard error (in brackets) values for different frequency bands in the contrast: High tinnitus-related distress (HD)** *versus* **low tinnitus-related distress (LD) group, calculated in each cluster separately for the Body Focus Condition (BFC) and Tinnitus Focus Condition (TFC).**
(PDF)

**S6 Table.** *Post-hoc* **pairwise *t*-tests results for the contrast: High tinnitus-related distress (HD)** *versus* **low tinnitus-related distress (LD) group, calculated in each cluster separately for the Body Focus Condition (BFC) and Tinnitus Focus Condition (TFC).** Items in bold are significant based on *p*-values.
(PDF)

## Acknowledgments

The authors are grateful to Andrew Bell for English language editing and valuable comments on the paper.

## Author Contributions

**Conceptualization:** Monika Lewandowska.

**Data curation:** Rafał Milner, Małgorzata Ganc.

**Formal analysis:** Rafał Milner, Monika Lewandowska, Jan Nikadon.

**Funding acquisition:** Monika Lewandowska, Henryk Skarżyński.

**Investigation:** Rafał Milner.

**Methodology:** Rafał Milner, Monika Lewandowska.

**Project administration:** Rafał Milner, Małgorzata Ganc, Iwona Niedziałek.

**Resources:** Iwona Niedziałek.

**Supervision:** Wiesław Wiktor Jędrzejczak, Henryk Skarżyński.

**Visualization:** Rafał Milner.

**Writing – original draft:** Rafał Milner, Monika Lewandowska.

**Writing – review & editing:** Rafał Milner, Monika Lewandowska.

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
