## [Decision Letter · Decision Letter 0]

1 May 2020

PONE-D-20-06961

Electrophysiological correlates of focused attention on low- and high-distressed tinnitus.

PLOS ONE

Dear Dr. Milner,

Thank you for submitting your manuscript to PLOS ONE. After careful consideration, we feel that it has merit but does not fully meet PLOS ONE’s publication criteria as it currently stands. Therefore, we invite you to submit a revised version of the manuscript that addresses the points raised during the review process.

It has been unusually difficult to find a second reviewer, but I decided to carry on because reviewer#1 has provided excellent comments that will help you to improve the final version of your interesting manuscript. I am sure that addressing his/her comments will make your manuscript clearer and stronger.

We would appreciate receiving your revised manuscript by Jun 15 2020 11:59PM. To enhance the reproducibility of your results, we recommend that if applicable you deposit your laboratory protocols in protocols.io, where a protocol can be assigned its own identifier (DOI) such that it can be cited independently in the future. For instructions see: http://journals.plos.org/plosone/s/submission-guidelines#loc-laboratory-protocols

We look forward to receiving your revised manuscript.

Kind regards,

Manuel S. Malmierca, M.D.,Ph.D.

Academic Editor

PLOS ONE

Journal Requirements:

Reviewers' comments:

Reviewer's Responses to Questions

**Comments to the Author**

1. Is the manuscript technically sound, and do the data support the conclusions?

Reviewer #1: Yes

2. Has the statistical analysis been performed appropriately and rigorously? 

Reviewer #1: Yes

3. Have the authors made all data underlying the findings in their manuscript fully available?

Reviewer #1: Yes

4. Is the manuscript presented in an intelligible fashion and written in standard English?

Reviewer #1: Yes

5. Review Comments to the Author

Reviewer #1: General:

This study addresses resting-state EEG in humans subjects with tinnitus, which has been an area of substantial interest and influence in the field of tinnitus neuroscience for at least 2 decades. Unfortunately this field has almost entirely neglected to control for some factors that might well impact greatly on ongoing brain activity, namely hearing loss and, particularly, attention. This study is one of only two to explicitly control and measure the correlates of attention in this context, and is the only one to include the all-important condition of diverting attention away from tinnitus.

Some general notes are below, followed by a moderate number of suggestions that might improve the quality of results and impact of the findings. However, nothing is ‘wrong’ as such with the manuscript in its present form, so I have generally kept these as optional.

Good sample size

Clear novelty over the Neff study (62), and is the first study to contrast attention toward vs away from tinnitus. This is a very important area of resting-state brain activity research in tinnitus, which until the present study and (62) has been totally neglected in the past 2 decades of this line of research.

Quality of writing is satisfactory

Introduction is generally factual, and avoids unwarranted speculation

Subjects well-matched for age, sex, tinnitus variables and hearing

Appropriate and well-performed methods. Appropriate use of statistical tests, and control for multiple comparisons.

The discussion is categorised by finding, but within those categories is rather sprawling and vague, discussing many possibilities for each finding, and arguing in favour of none specifically. However, I do not think there is necessarily anything wrong with this, and it is definitely preferable to confidently stating unjustified conclusions. The discussion in its present form does speculate a lot in many places, including into some rather tenuous explanations, but in every case it is clearly presented as speculation, and therefore I think that is, once again, fine.

Major:

There is little mention made of how the brain changes associated with tinnitus distress compare to those associated with other chronic aversive sensory conditions, such as chronic pain, on which there is a substantial literature. When I last looked at this, albeit some years ago, there were striking parallels. I do not feel strongly that a discussion of this should necessarily be added to the manuscript, as it would require significant work, but if included I think it would make the manuscript considerably more interesting and useful, as it would help determine the extent to which the observed results are generic features of distress or chronic symptoms (more likely) or specific to tinnitus (less likely).

Was any analysis done on gamma band activity? There has been a lot of interest in resting-state EEG gamma in tinnitus, and it would be useful to know whether the present findings can contribute to this ongoing area of interest.

It is a shame that all analysis has been performed in sensor space as opposed to source space. As there is such a loose correspondence between specific electrodes and brain areas, this means that it is hard to compare the present findings with most of the previous literature, which was presented in source space, mainly using sLORETA. Again this is optional (PLoS One only requires the experiments to be properly conducted and reported), but the findings would be so very much more useful, interesting and impactful if an sLORETA analysis were performed. The sLORETA software is very easy and quick to use, free, and accepts data in a .txt file format following simple conventions such that it can easily accept preprocessed data from other software packages. Reanalysis with sLORETA should take less than a week, and would add so much. But, this is the authors’ decision.

I rarely say this in peer reviews, but I wonder if the authors’ analysis has been overly conservative, in that I do not think depression and anxiety scores necessarily need to be factored out when conducting the statistical analyses. Yes, they are correlated to tinnitus distress, but unless the subjects had pre-existing mental health conditions prior to tinnitus then these scores are probably consequences of tinnitus distress (or at least an inherent part of the mechanism of tinnitus distress), and hence factoring them out will probably remove a lot of variance from the data that genuinely reflects the variable of interest (tinnitus distress). Other variables (age, sex, hearing, duration, ear) did not vary significantly between groups, and therefore I would be satisfied to just leave it at that and not address them in the analysis. With this in mind, I am not sure the PCA-based analysis is necessary, but I will leave this to the authors’ discretion.

I also wonder (though am not sure) if the use of ANOVA here has applied too harsh a statistical penalty. We must bear in mind that the different ROIs, and different frequency bands, are far from independent. This is because many effects are common across multiple frequency bands, and most brain sources project much more widely to the scalp than just one of the authors’ ROIs as used here. I believe that the ANOVA will assume independence of frequencies and ROIs, and therefore its results will be much too conservative. In situations such as these, I tend to base all my stats on non-parametric permutation testing (i.e. run the analysis, say, 1000 times over but randomising the subject group labels each time, and taking the maximum group difference observed in each permutation to construct a null distribution, and finally use the 0.05 position in the sorted null distribution as the significance threshold for the real data). This technique works across all types of data, makes no assumptions about distributions, corrects appropriately for multiple comparisons, and is well-suited to large multivariate datasets. Once again, I do not feel strongly that the authors do this in this case, but if they choose not to then it should be made clear how conservative the results are likely to have been as a result of the methods chosen, and how this impacts on the interpretation of any negative results in particular frequency bands and/or brain locations. Note that sLORETA has this method, or an equivalent, built into its functionality, should the authors decide to follow my recommendation to perform a source space analysis.

Related to my comment in the ‘general’ section, regarding the discussion. Given the lack of a clear unifying interpretation of the findings, the authors might consider adopting a ‘less is more’ approach, and trying to keep the discussion brief and tidy, as the huge number of potential, and potentially conflicting, speculative interpretations in the present version is hard to derive any kind of take-home message from, and therefore does not really give any additional value to the reader. However, it is the authors’ right to speculate as much as they wish, and I will not insist on this section being changed.

Minor:

Two successive sentences in the abstract describe results with and without different degrees of correction, and appear to contradict each other. I would advise sticking to just one analysis, hence one set of findings, in the abstract. As I argue in ‘major’, I would for once advocate focusing on the less conservative analysis.

In the introduction, discussion of the resting-state EEG literature does not mention that all these prior studies failed to control for hearing loss (and sometimes even for age), and omits Adjamian et al. (2012 – JARO) which is rare in actually controlling for hearing loss (and did not replicate most of the previously reported difference).

The introduction presents the thalamocortical dysrhythmia theory as if it is accepted fact. I would suggest stating it as a possible theory, and also perhaps acknowledging that the studies supporting it did not control for hearing loss (or attention!), and likewise that they only recorded from cortex, to the link to thalamus is speculative.

The statement “probably due to impaired habituation” should be rephrased, as by definition what the authors are describing is impaired habituation.

The statement “whereas less persistent tinnitus is associated”: the authors presumably mean ‘less bothersome’, or noticeable a smaller proportion of the time.

Line 102 should read “… is related to…”

129: “highly” not “high”

135: why “paradoxical”? It seems intuitive that this should be the case

142: “suggest” not “confirm”

145: “correlates of their EEG” does not make sense. Do the authors mean “EEG correlates of tinnitus-directed attention” or similar?

Lines 148-151: The rationale is not made clear for why the present study should provide insights not already provided by the Neff study (62). In my mind, the novelties of the present study are the stratification by distress level, and the use of an ‘attend-away’ condition (the body-focused condition). In principle, there might not be much or any difference between attending to tinnitus vs. resting state if subjects are already attending to their tinnitus in resting state, whereas a condition requiring attention away from tinnitus might be more likely to reveal differences.

Was any verification made, after the experiment, of how much subjects adhered to the instructed focus of attention for each block (and whether they had their eyes open or closed)?

Where statistical results are reported, are the authors able to provide p values? Readers will find these the easiest measures to gain a quantitative impression of. I note there are p values in the figures, but it would be nice to have them in the text also.

Line 504: I wouldn’t make too much of ref (39), because in hindsight the results almost certainly just indicated an excess of jaw muscle artefact in the tinnitus group

My PDF file did not seem to include any captions for the main text figures. I assume that these are present somewhere.

6. PLOS authors have the option to publish the peer review history of their article (what does this mean?). If published, this will include your full peer review and any attached files.

Reviewer #1: Yes: William Sedley

---

## [Author Response · Author response to Decision Letter 0]

2 Jul 2020

We would like to thank the Reviewer for thorough reading of our manuscript and for kind comments and constructive suggestions. We really appreciate all of them and consider as very helpful for improving the manuscript. We agree with all comments and we have revised the manuscript accordingly. We responded to the comments and suggestions chronologically, beginning from the major remarks. Our responses follow the Reviewer’s comments (in italics). 

Major remarks

“There is little mention made of how the brain changes associated with tinnitus distress compare to those associated with other chronic aversive sensory conditions, such as chronic pain, on which there is a substantial literature. When I last looked at this, albeit some years ago, there were striking parallels. I do not feel strongly that a discussion of this should necessarily be added to the manuscript, as it would require significant work, but if included I think it would make the manuscript considerably more interesting and useful, as it would help determine the extent to which the observed results are generic features of distress or chronic symptoms (more likely) or specific to tinnitus (less likely).”

Replay: 

Thank you for this suggestion. Since we agree that tinnitus and other aversive sensory conditions (especially chronic pain) have a lot in common, we’ve added some paragraphs addressing this issue to the Discussion section (page 20; line 459-469 – marked-up copy of manuscript).

“Was any analysis done on gamma band activity? There has been a lot of interest in resting-state EEG gamma in tinnitus, and it would be useful to know whether the present findings can contribute to this ongoing area of interest.”

Replay: 

Yes, the current version of the manuscript contains the outcomes of both EEG power and current density (sLORETA) analysis for 8 frequency bands (including gamma). 

„It is a shame that all analysis has been performed in sensor space as opposed to source space. As there is such a loose correspondence between specific electrodes and brain areas, this means that it is hard to compare the present findings with most of the previous literature, which was presented in source space, mainly using sLORETA. Again this is optional (PLoS One only requires the experiments to be properly conducted and reported), but the findings would be so very much more useful, interesting and impactful if an sLORETA analysis were performed. The sLORETA software is very easy and quick to use, free, and accepts data in a .txt file format following simple conventions such that it can easily accept preprocessed data from other software packages. Reanalysis with sLORETA should take less than a week, and would add so much. But, this is the authors’ decision.”

Replay:

In the “Limitations of the study” section in previous version of the manuscript we’ve mentioned about calculating LORETA and presenting the results in our next work. However, following the suggestion, we have added the outcomes of current density analysis to this manuscript. At the same time, because after re-computing EEG power data, some new interesting effects came up, we’ve decided to include the results of both analyses, in sensor and source space.

“I rarely say this in peer reviews, but I wonder if the authors’ analysis has been overly conservative, in that I do not think depression and anxiety scores necessarily need to be factored out when conducting the statistical analyses. Yes, they are correlated to tinnitus distress, but unless the subjects had pre-existing mental health conditions prior to tinnitus then these scores are probably consequences of tinnitus distress (or at least an inherent part of the mechanism of tinnitus distress), and hence factoring them out will probably remove a lot of variance from the data that genuinely reflects the variable of interest (tinnitus distress). Other variables (age, sex, hearing, duration, ear) did not vary significantly between groups, and therefore I would be satisfied to just leave it at that and not address them in the analysis. With this in mind, I am not sure the PCA-based analysis is necessary, but I will leave this to the authors’ discretion.”

Replay: 

Actually, those of us who are clinicians working with tinnitus patients, are convinced that when examining their brain activity, it is always worth to consider the tinnitus characteristics (etiology, duration, persistence, etc.) and accompanying symptoms (in addition to the results of audiological tests, the measures of depression and anxiety levels, an ability to cope with stress, etc.). It happens that these psychological measures are more useful for proper diagnosis and treatment than just the subjectively-reported tinnitus intrusiveness (by the way a psychological interview would be more informative than any questionnaires). However, we agree that in this study, where the compared groups are well matched in terms of demographic variables (sex and age), the audiogram profile (up to 8 kHz) and tinnitus duration, as well as the anxiety and depression levels in the whole sample are relatively low (suggesting that they are simply a consequence of tinnitus distress) it is not necessary to include all these variables into the statistical model. Therefore, we decided to remove from the manuscript the paragraphs concerning the data computations when all these factors were controlled. We also gave up the results of PCA-based analysis. 

“I also wonder (though am not sure) if the use of ANOVA here has applied too harsh a statistical penalty. We must bear in mind that the different ROIs, and different frequency bands, are far from independent. This is because many effects are common across multiple frequency bands, and most brain sources project much more widely to the scalp than just one of the authors’ ROIs as used here. I believe that the ANOVA will assume independence of frequencies and ROIs, and therefore its results will be much too conservative. In situations such as these, I tend to base all my stats on non-parametric permutation testing (i.e. run the analysis, say, 1000 times over but randomising the subject group labels each time, and taking the maximum group difference observed in each permutation to construct a null distribution, and finally use the 0.05 position in the sorted null distribution as the significance threshold for the real data). This technique works across all types of data, makes no assumptions about distributions, corrects appropriately for multiple comparisons, and is well-suited to large multivariate datasets. Once again, I do not feel strongly that the authors do this in this case, but if they choose not to then it should be made clear how conservative the results are likely to have been as a result of the methods chosen, and how this impacts on the interpretation of any negative results in particular frequency bands and/or brain locations. Note that sLORETA has this method, or an equivalent, built into its functionality, should the authors decide to follow my recommendation to perform a source space analysis.”

Replay: 

Thank you for this suggestion. In fact while analyzing the EEG data we had a feeling that we could get more effects from our dataset than we reported in the previous version of the manuscript. Therefore, intending to be not very far from the analysis that we did last time, we decided to calculate repeated measures ANOVA on QEEG data and also provide p-values based on the permutation methods that handle nuisance variables. This approach allows to reveal some new significant effects that also need to be discussed (please find the parts of the Results and Discussion Sections written in bold). Following the recommendation we also added sLORETA outcomes to the manuscript. 

“Related to my comment in the ‘general’ section, regarding the discussion. Given the lack of a clear unifying interpretation of the findings, the authors might consider adopting a ‘less is more’ approach, and trying to keep the discussion brief and tidy, as the huge number of potential, and potentially conflicting, speculative interpretations in the present version is hard to derive any kind of take-home message from, and therefore does not really give any additional value to the reader. However, it is the authors’ right to speculate as much as they wish, and I will not insist on this section being changed.”

Replay:

In the current version of the manuscript we’ve done our best to make the Discussion section more clear and concise. We've also put much effort to, wherever it was possible, avoid speculative interpretations. The “Conclusions” was added to provide a take-home message.

Minor:

„Two successive sentences in the abstract describe results with and without different degrees of correction, and appear to contradict each other. I would advise sticking to just one analysis, hence one set of findings, in the abstract. As I argue in ‘major’, I would for once advocate focusing on the less conservative analysis.”

Since we decided to re-compute the data, the abstract was also re-written accordingly. 

“In the introduction, discussion of the resting-state EEG literature does not mention that all these prior studies failed to control for hearing loss (and sometimes even for age), and omits Adjamian et al. (2012 – JARO) which is rare in actually controlling for hearing loss (and did not replicate most of the previously reported difference).”

Thank you for pointed out this issue. We added a sentence about this to Introduction (page 4; lines 89-90 – marked-up copy of manuscript).

“The introduction presents the thalamocortical dysrhythmia theory as if it is accepted fact. I would suggest stating it as a possible theory, and also perhaps acknowledging that the studies supporting it did not control for hearing loss (or attention!), and likewise that they only recorded from cortex, to the link to thalamus is speculative.”

We are sorry if we’ve made an impression that we consider the thalamocortical dysrhythmia theory as widely accepted. We are aware of the fact that this model is only one possible among others (we’ve already mentioned about them in Introduction, lines 75-78). Therefore, we corrected a little bit a paragraph that covers the theories explaining tinnitus (lines 66-78) and also a sentence elsewhere in the text (lines 83-84). 

“The statement “probably due to impaired habituation” should be rephrased, as by definition what the authors are describing is impaired habituation”.

Thank you. We’ve corrected this sentence (lines 91-92).

“The statement “whereas less persistent tinnitus is associated”: the authors presumably mean ‘less bothersome’, or noticeable a smaller proportion of the time.”

Yes, of course we meant “less bothersome” not “less persistent” tinnitus. Thank you for catching this mistake. 

“Line 102 should read “… is related to…

The suggested correction has been made.

“129: “highly” not “high””. 

This has been corrected (line 137).

“135: why “paradoxical”? It seems intuitive that this should be the case”

The “paradoxical” has been removed from this sentence. In this context this word truly makes no sense. 

“142: “suggest” not “confirm””

This correction has been made (line 150).

“145: “correlates of their EEG” does not make sense. Do the authors mean “EEG correlates of tinnitus-directed attention” or similar?”

Thank you, we fully agree with that. This sentence has been re-written (line 152).

“Lines 148-151: The rationale is not made clear for why the present study should provide insights not already provided by the Neff study (62). In my mind, the novelties of the present study are the stratification by distress level, and the use of an ‘attend-away’ condition (the body-focused condition). In principle, there might not be much or any difference between attending to tinnitus vs. resting state if subjects are already attending to their tinnitus in resting state, whereas a condition requiring attention away from tinnitus might be more likely to reveal differences.”

Thank you for bringing up this important issue. Resting-state brain activity is defined as activity in the brain when a subject is awake but not performing a specific cognitive task or responding to sensory stimuli (e.g. Fox & Greicius, 2010). Indeed, in this context in the present study EEG data were recorded during the resting-state (understood as the absence of overt task performance or stimulation). It is also true that we expected to observe the most pronounced differences in the brain activity of high distress patients between the conditions when they were attending to, or not, to their tinnitus. We are not, however, convinced that a state of concentrating on tinnitus is the same as the “resting state” (even in bothersome tinnitus), especially when, as it was in the current study, patients were asked to keep their eyes closed (this condition favors focusing on the inner world). We believe that in our study a way of formulating an instruction given to a patient, actually mattered. Typically when we measure spontaneous brain activity (with the use of EEG or fMRI) we ask a subject to sit (or lie) with eyes open, eyes fixated on a target or eyes closed, and relax, not thinking about anything special. We have had some experience in conducting resting-state studies either with the use of EEG or fMRI technique, and it is almost impossible to request subjects only to keep their eyes open or closed because they often ask what exactly they are supposed to do during the experiment. While preparing the current study we were concerned that such instruction might be misunderstood, especially by patients with high tinnitus-related distress. Specifically they might have found this instruction encouraging not to focus on tinnitus (which could have been related to much cognitive effort reflected in EEG activity). Therefore, we decided to compare brain activity between more “controlled” resting-state conditions.

“Was any verification made, after the experiment, of how much subjects adhered to the instructed focus of attention for each block (and whether they had their eyes open or closed)?”

Our participants kept their eyes closed during both blocks of recording EEG signal (line 231). We did not mention about it in the previous version of the manuscript but immediately after completing each block all patients were requested to assess how well they had been able to follow the instruction on the 5-point scale from “1” (completely failed) to “5” (completely done). Only those persons who reported “5” or “4” (almost completely done) in both conditions were included to the study. However, since only few persons admitted that they failed to follow the instruction, we were not sure whether other declarations were true and, therefore, we finally decided to not insert this information into the manuscript (and we would like to keep it that way). In future studies we should definitely consider a better verification procedure of the instruction adherence. 

“Where statistical results are reported, are the authors able to provide p values? Readers will find these the easiest measures to gain a quantitative impression of. I note there are p values in the figures, but it would be nice to have them in the text also.”

In the current version of the manuscript p values were included in both figures and in the text (please note this in the Results section). 

“Line 504: I wouldn’t make too much of ref (39), because in hindsight the results almost certainly just indicated an excess of jaw muscle artefact in the tinnitus group”

We take this hint into consideration and we did not discuss our results with the reference to this paper in the he current version of the manuscript. 

“My PDF file did not seem to include any captions for the main text figures. I assume that these are present somewhere.”.

We are very sorry for that. We hope that in the corrected manuscript the captions for all figures are in the right place. In accordance with the Plos One guidelines, we included them in the text of the manuscript. 

References

Fox MD, Greicius M. Clinical applications of resting state functional connectivity. Front Syst Neurosci. 2010;4:19. Published 2010 Jun 17. doi:10.3389/fnsys.2010.00019

---

## [Decision Letter · Decision Letter 1]

9 Jul 2020

Electrophysiological correlates of focused attention on low- and high-distressed tinnitus.

PONE-D-20-06961R1

Dear Dr. Milner,

We’re pleased to inform you that your manuscript has been judged scientifically suitable for publication and will be formally accepted for publication once it meets all outstanding technical requirements.

Kind regards,

Manuel S. Malmierca, M.D.,Ph.D.

Academic Editor

PLOS ONE

Additional Editor Comments (optional):

Reviewers' comments:

Reviewer's Responses to Questions

**Comments to the Author**

1. If the authors have adequately addressed your comments raised in a previous round of review and you feel that this manuscript is now acceptable for publication, you may indicate that here to bypass the “Comments to the Author” section, enter your conflict of interest statement in the “Confidential to Editor” section, and submit your "Accept" recommendation.

Reviewer #1: All comments have been addressed

2. Is the manuscript technically sound, and do the data support the conclusions?

Reviewer #1: Yes

3. Has the statistical analysis been performed appropriately and rigorously? 

Reviewer #1: Yes

4. Have the authors made all data underlying the findings in their manuscript fully available?

Reviewer #1: Yes

5. Is the manuscript presented in an intelligible fashion and written in standard English?

Reviewer #1: Yes

6. Review Comments to the Author

Reviewer #1: I really appreciate the authors' careful consideration of all my points, and the substantial work and effort they have undergone in order to fully embrace and address these.

The new version of the manuscript is much improved by the additional source space analysis, and more relaxed statistical analyses, and the results that these have produced. I also note the numerous instances of rewording in the manuscript which has helped it read in a more even-handed way (though this was never much of an issue to begin with).

The synthesis of results, and discussion, seem to deliver more of a clear take-home message too.

Overall, I continue to like this manuscript, and think it has been greatly further improved by the additional work and effort. I am enthusiastic to recommend publication in its present form.

7. PLOS authors have the option to publish the peer review history of their article (what does this mean?). If published, this will include your full peer review and any attached files.

Reviewer #1: **Yes: **William Sedley

---

## [Editor Report · Acceptance letter]

17 Jul 2020

PONE-D-20-06961R1 

Electrophysiological correlates of focused attention on low- and high-distressed tinnitus. 

Dear Dr. Milner:

I'm pleased to inform you that your manuscript has been deemed suitable for publication in PLOS ONE. Congratulations! Your manuscript is now with our production department. 

Kind regards, 

on behalf of

Prof. Dr. Manuel S. Malmierca 

Academic Editor

PLOS ONE